# New Benthic Cyanobacteria from Guadeloupe Mangroves as Producers of Antimicrobials

**DOI:** 10.3390/md18010016

**Published:** 2019-12-23

**Authors:** Sébastien Duperron, Mehdi A. Beniddir, Sylvain Durand, Arlette Longeon, Charlotte Duval, Olivier Gros, Cécile Bernard, Marie-Lise Bourguet-Kondracki

**Affiliations:** 1Molécules de Communication et Adaptation des Microorganismes, UMR 7245 CNRS, Muséum National d’Histoire Naturelle, 57 rue Cuvier (CP54), 75005 Paris, Francearlette.longeon@mnhn.fr (A.L.); charlotte.duval@mnhn.fr (C.D.); cecile.bernard@mnhn.fr (C.B.); 2Institut Universitaire de France, 75005 Paris, France; 3Équipe “Pharmacognosie-Chimie des Substances Naturelles” BioCIS, CNRS, Université Paris-Saclay 5 rue Jean-Baptiste Clément, 92290 Châtenay-Malabry, France; mehdi.beniddir@u-psud.fr; 4UMR 7205 ISYEB et Université des Antilles, Pointe à Pitre, 97157 Guadeloupe, France; olivier.gros@univ-antilles.fr

**Keywords:** benthic cyanobacteria, tropical mangrove, Guadeloupe, phylogenetic diversity, chemical diversity, molecular networking, antimicrobial activity

## Abstract

Benthic cyanobacteria strains from Guadeloupe have been investigated for the first time by combining phylogenetic, chemical and biological studies in order to better understand the taxonomic and chemical diversity as well as the biological activities of these cyanobacteria through the effect of their specialized metabolites. Therefore, in addition to the construction of the phylogenetic tree, indicating the presence of 12 potentially new species, an LC-MS/MS data analysis workflow was applied to provide an overview on chemical diversity of 20 cyanobacterial extracts, which was linked to antimicrobial activities evaluation against human pathogenic and ichtyopathogenic environmental strains.

## 1. Introduction

Cyanobacteria are among the important primary producers in various coastal ecosystems including mangroves. Besides their occurrence in the bacterioplankton, various cyanobacteria occur in biofilms on the sediment surface, on rocks, and on biological surfaces as part of the periphyton [1,2]. Biofilm-forming cyanobacteria contribute to locale trophic networks through carbon fixation, and depending on species also to nitrogen fixation, accumulation of calcium, magnesium, and phosphorous [3]. The benthic species, especially in tropical zones, may form dense biofilms on various types of substrates and may have major ecological roles [2] but are still poorly known, compared to pelagic species. Although cyanobacteria are of particular interest as ecologically-relevant microorganisms, they are also regarded as producers of a broad diversity of bioactive secondary metabolites including cyanotoxins and various antimicrobial compounds which influence their interactions with other organisms [1,4,5]. Some of these compounds are of pharmacological interest, as illustrated by the use of Brentuxymab vedotin, based on dolastatin 10 from *Symploca*, in the treatment of Hodgkin’s lymphoma [1,6]. 

Currently, the 1700 described cyanobacterial species [7] are probably only a small subset of the group’s true diversity. The tropical regions and the benthic compartment are particularly ill-explored compared to the potential diversity their harbor [2]. Chemical investigations have focused on an even smaller subset of this diversity, with over 50% of characterized metabolites reported from the order Oscillatoriales, and 35% in the sole genus *Lyngbya* [5].

In this study, we investigated both the phylogenetic and chemical diversity of cultivable cyanobacteria isolated from coastal habitats in Guadeloupe (French West Indies). Marine benthic cyanobacteria from Guadeloupe have indeed received very little attention despite the fact that they can form biofilms that may cover large areas of sediment and plant surfaces. Three filamentous morphotypes were, for example, recently characterized, but were not maintained in culture collections [8]. In the present study, 20 bacterial strains were isolated from various biofilms in Guadeloupe collected from the benthic sediment surface or from the surface of immersed mangrove tree branches and roots in the Manche-à-Eau mangrove lagoon, the Marina Bas-du-Fort harbor, and the Canal Des Rotours, a 6-km long canal connecting the city of Morne-à-l’eau to the coastal mangrove. Strains were characterized by means of 16S rRNA comparative gene sequence analysis, and their metabolites were analyzed using LC-MS/MS. A molecular network was built to establish chemical entities families, which were compared among strains [9]. Finally, the antibacterial activities were evaluated against six human and four marine pathogen reference strains. This study provides a first glimpse of the taxonomic and chemical diversity of the benthic cyanobacteria occurring in Guadeloupe coastal mangroves.

## 2. Results 

A total of 20 cyanobacterial strains were successfully isolated, and grown from green biofilms collected from distinct locations in Guadeloupe, namely the Manche-à-Eau lagoon close to red mangrove trees (two stations ST1 and ST2 and five strains), the Marina Bas-du-Fort (one station ST4, one strain), and the Canal Des Rotours (three stations ST5, ST6, and ST7 and 14 strains, Figure 1 and Table 1). Biofilms were found either covering the sediment surface, attached to immersed roots of mangrove trees, or attached to sunken deadwood.

### 2.1. Phylogenetic and 16S rRNA Dissimilarity-Based Identification of Strains

The 16S rRNA-encoding gene sequences from the 20 strains clustered within the three cyanobacterial orders Oscillatoriales, Synechococcales and Nostocales (ten, eight, and two strains, respectively, Figure 2). Strains were affiliated to hypothetical species and genera based on widely accepted 16S rRNA similarity cutoff values for species and genus delimitation [7,10], respectively, and monophyly with members of these species and genera. A large-scale comparison of 6787 genomes from 22 prokaryotic phyla established that a 99% 16S rRNA similarity cutoff value should be retained to delimit species [10]; and reference taxonomic works on Cyanobacteria recommend a 95% cutoff of cyanobacterial genera delimitation [7].

Five of the ten Oscillatoriales strains clustered with sequences from uncultured *Oscillatoria* and a strain assigned to *Kamptonema formosum* (formely *Oscillatoria formosa*). Using the aforementioned criteria, these strains are congeneric, and can be assigned to three new hypothetical species represented by strain PMC 1075.18, strains PMC 1068.18 and 1076.18, and strains PMC 1050.18 and 1051.18, respectively (Table 1 and Appendix A). Strains PMC 1056.18 and 1057.18 (0.8% dissimilarity) from Manche-à-Eau represent one hypothetical new species that displays 4% dissimilarity with the closest described genus, *Arthrospira*, and thus probably belong to this genus. Strains PMC 1071.18 (7.8% dissimilarity with *Symploca* sp. NAC 12/21/08-3), and PMC 1072.18 (5.1% dissimilarity with *Ramsaria avicennae* and *Coleofasciculus chthonoplastes*) represent two new species, each belonging to a new genus based on the 95% similarity threshold. Strain PMC 1092.19 displayed 2.2% dissimilarity with *Lyngbya* sp. ALCB114379 and thus likely represents a new species within this genus. Within the Synechococcales, five strains (PMC 1073.18, 1074.18, 1078.18, 1079.18, and 1080.18, all from Canal des Rotours) clustered with genus *Jaaginema* and represent a single new hypothetical species. Strain PMC 1066.18 was closely related (0.5% sequence dissimilarity) to a sequence from an uncultured *Nodosilinea* sp. CENA 322, isolated from leaves of the mangrove tree *Avicennia schaueriana* in Brazil [11]. Sequence from strain PMC 1064.18 was highly similar (0.5% dissimilarity) to sequences from two unpublished strains of *Limnothrix*. Strain PMC 1052.18 displayed at least 7.4% dissimilarity to all other sequences in databases, and represents a new species within a new undescribed genus. Finally, strains PMC 1069.18 and 1070.18 from Canal des Rotours were the only two Nostocales, clustering together and representing a single new hypothetical species belonging to the genus *Scytonema.*

### 2.2. MS/MS Analysis and Annotation of Cyanobacterial Specialized Metabolites

In an attempt to map the chemical diversity of the 20 cyanobacterial extracts, their LC-MS/MS data were preprocessed using MZmine 2 [12] and the obtained 2468 mass features were organized into a molecular network consisting of 156 clusters (two or more connected nodes of a graph, Figure 3). In order to visualize the distribution of the cyanobacterial metabolites across the 20 extracts, the whole molecular network was mapped at the genus identification level using a typical color tag (Figure 3). An examination of the network reveals that certain clusters are constrained to specific genera. This observation highlighted the distribution of closely related yet different chemical structures in each genus (See Appendix A for further details). Moreover, MS/MS data constituting the entire molecular network were searched against the GNPS spectral libraries [13] and yielded only 9 hits (triangle shapes on the network), including a nucleotide, diketopiperazines and phosphocholines (https://gnps.ucsd.edu/ProteoSAFe/result.jsp?task=9581427a15b7422d8bd2b3b4b086189e&view=view_all_annotations_DB). To further expand the annotation coverage, we applied DEREPLICATOR, a recently developed dereplication algorithm that enables high-throughput peptide natural products (PNPs) identification from their tandem mass spectra [14,15,16,17]. Interestingly, this tool allows to putatively identify an unknown PNP from its known variants (for example, with a substitution, a modification or an adduct) through the so-called variable dereplication process (as opposed to the strict dereplication when a PNP is described in the database). This dereplication process allowed the annotation of 54 peptide spectrum matches (PSMs) (red ellipses on the network, Figure 3) analogues closely related to known ones (Appendix A). Even though, no perfect match was generated through DEREPLICATOR algorithm, marine peptide natural products were proposed and therefore support the relevance of this tool for the exploration of peptide diversity. Furthermore, on the basis of the taxonomic annotation, some clusters were restricted to a single genus indicating the chemical uniqueness of the features within the whole map and, potential structural originality (Appendix A) [18].

### 2.3. Evaluation of the Antimicrobial Activity 

The evaluation of the antimicrobial activity was performed against six human pathogenic bacteria (*Escherichia coli*, *Klebsiella pneumoniae*, *Pseudomonas aeruginosa*, *Enterococcus faecalis*, *Staphylococcus aureus,* and *Bacillus cereus*) and four marine environmental pathogenic bacteria (*Vibrio alginolyticus*, *Vibrio anguillarum*, *Pseudoalteromonas atlantica,* and *Pseudoalteromonas distincta*). All the cyanobacterial strains extracts were tested in triplicates at a concentration of 100 µg/mL. Only positive results (i.e., inhibition above 50%) are presented in Table 2.

Most of the cyanobacterial strains showed moderate activity against *P. atlantica* with a mean growth inhibition of 20% to 60%; three *Jaaginema* strains PM 1078.18, 1079.18 and 1080.18 as well as *Oscillatoria n. sp. 2* PMC 1076.18 displayed highest growth inhibition properties.

Interestingly, *Oscillatoria/Kamptonema* strain PMC 1051.18 was the only strain to reveal significant activity against *E. coli* with 100 % growth inhibition at 100 µg/mL, while its conspecific strain PMC 1050.18 did not show significant activity.

## 3. Discussion and Conclusions

### 3.1. Mangroves of Guadeloupe are a Source of Novel Cyanobacteria

This study yielded 20 new cyanobacterial strains including ten Oscillatoriales, eight Synechococcales and two Nostocales [7]. Based on 16S rRNA phylogeny and similarity criteria, these possibly represent 12 new species within three new and seven already known genera (*Jaaginema*, *Scytonema*, *Oscillatoria*, *Nodosilinea, Lyngbya, Arthrospira,* and *Limnothrix*). Ascertaining the exact status of strains as new species or genera requires further characterization beyond simple 16S rRNA dissimilarity- and phylogeny-based criteria considered here. A polyphasic approach is required in order to offer a proper taxonomic description, that is beyond the scope of this paper [7,19]. However, this result based on the analysis of 16S rRNA-encoding genes alone, already emphasizes the high level of taxonomic novelty found in Guadeloupe coastal environments. Apart from a recent paper describing three cyanobacteria not closely related to strains described herein, namely *Oscillatoria* sp. clone gwada strain OG (displaying over 5 % 16S rRNA sequence dissimilarity with our *Oscillatoria*/*Kamptonema* sequences), ‘*Candidatus* Planktothricoides niger’ strain OB and ‘*Candidatus Planktothricoides rosea*’ strain OP, very little data is available regarding cyanobacterial taxonomic diversity in Guadeloupe [8]. Overall the level of novelty documented here reveals mostly untapped cyanobacterial diversity in Guadeloupe. Various authors have highlighted the potential of tropical ecosystems, in particular mangroves, as reservoirs of cyanobacterial diversity and bioactive molecules [11,20], and the predominance of orders Oscillatoriales and Synechococcales among recovered strains [21,22], so this novelty is not unexpected and warrants further investigations. 

### 3.2. The Benthic Cyanobacterial Strains Reveal a High Level of Chemical Novelty

The presence of 54 peptides, variants of known previously identified peptides, was documented using the DEREPLICATOR algorithm. These latter include eleven peptides previously isolated from yanobacteria (Table 3 and Appendix A) but also twelve known peptides from marine Sponges, three from Chordata, five from Fungi (including two marine fungi), and ten from Bacteria, whose one marine bacterium (Appendix A). These results suggest that some of the previously isolated peptides from marine organisms could have a symbiotic origin and could be in fact produced by cyanobacteria associated with these organisms.

The comparison of ions observed in the chemical extracts of the 20 strains also revealed high heterogeneity in chemical composition despite that strains were grown under similar conditions (temperature, Z8 media, photoperiod). This indicates high levels of inter-strain variability. 

The assessment of those cyanobacterial extracts using molecular networking and in silico annotation tool finally pointed out a high level of chemical novelty. Notably, this endeavor will allow prioritizing cyanobacterial strains for further chemical studies that will be pursued with full structure elucidation of each isolated compound through detailed NMR analyses.

### 3.3. Certain Cyanobacterial Strains Reveal Promising Antimicrobial Activities 

All strains inhibited the growth of *P. atlantica* to a certain extent. The most active strains, namely PMC 1078.18, 1079.18, and 1080.18, displaying more than 50% of inhibitory activity, belong to a single hypothetical new species of the genus *Jaaginema* and were isolated from the Canal des Rotours. The two other conspecific strains PMC 1073.18 and 1074.18 did not display a significant inhibition, suggesting that this property is limited to certain strains within this species. More interestingly, a single strain of *Oscillatoria*/*Kamptonema*, namely PMC 1051.18, isolated in the Manche à Eau lagoon, showed a very high inhibition against the human pathogenic bacterium strain *E. coli* with 100% of inhibition at the concentration of 100 µg/mL. This promising activity stimulates a deeper chemical study of this strain in order to isolate and identify the molecule(s) responsible for the detected activity. The antimicrobial activity assay will also be enlarged with assays against resistant pathogenic bacterial strains. Surprisingly, its closest relative in the phylogenetic tree, strain PMC 1050.18 did not display such an activity, again suggesting a strain-specific level of activity. 

Altogether these findings support that activities are limited to certain strains of a given cyanobacterial species. Inter-strain variability is most often attributed to the existence of certain genes or pathways that are found in some, but not all strains of a given species [23]. This could explain some of the chemical and activity differences observed herein between closely related strains. Alternatively, recent studies on cyanobacterial strains maintained in culture have revealed the existence of an overlooked associated cyanosphere, i.e., a cohort of other microorganisms (mostly bacteria) that is co-isolated and co-cultured with the cyanobacterium [24]. In this case, as observed in other organisms that co-exist with a microbiota, this cyanosphere certainly interacts and influences the cyanobacterial strains physiology, possibly being in obligate symbiosis, a consequence being the reported difficulty to obtain axenic cultures in Cyanobacteria [24]. These interactions might result in strain-specific differences in chemical composition and activities, either by modulating cyanobacterial gene expression, or because compounds and activities are in fact due to other members of the cyanosphere, and not the cyanobacterium itself. This may explain why different strains within a single species and cultured under similar conditions can display very different compounds and properties, and emphasizes the necessity to account for this heterogeneity by investigating various strains in parallel within each species.

In conclusion, easy-to-sample coastal areas in Guadeloupe harbor an untapped diversity of benthic cyanobacteria that probably represent novel lineages and display a diversity of potentially novel molecules, some of which have promising antimicrobial properties. The taxonomic affiliation of the strains has to be further investigated using polyphasic approaches (that include morphological, ultrastructural and molecular analyses). The different types of activities of the isolated peptides have also to be further explored. It certainly emphasizes the need to further investigate the different habitats in Guadeloupe, in particular mangroves, and more generally tropical coastal habitats, as environmental conditions have a major impact on cyanobacterial diversity [20]. To understand the significance of these Cyanobacteria in these ecosystems, and the possible roles of antibacterial compounds in nature, work is under way to address the actual abundance and functions of isolated strains in biofilms in natura using additional approaches including metagenomics, metabolomics and metatranscriptomics. This study paves the way for further promising investigations on benthic cyanobacteria from tropical mangroves.

## 4. Materials and Methods 

### 4.1. Sampling and Strain Isolation 

Green biofilms, presumed to contain abundant cyanobacteria, were sampled in July 2018 from distinct locations in Guadeloupe, namely two stations located in the Manche-à-Eau lagoon close to red mangrove trees (*Rhizophora mangle,* [25], five strains), the Marina Bas-du-Fort (one strain), and the Canal Des Rotours, a 6-km long canal build between 1826 and 1830 connecting the city of Morne-à-l’eau to the sea on three stations, representing a transition between the coastal mangrove (stations 5 and 6 with six and three strains, respectively) and freshwater (station 7 with five strains, Table 1). In accordance with Article 17, paragraph 2, of the Nagoya Protocol on Access and Benefit-sharing, a sampling permit was issued and published (https://absch.cbd.int/database/ABSCH-IRCC-FR-246959). Biofilms, either benthic or attached to submerged tree roots or branches (periphyton) were sampled by snorkeling. Back to the lab, biofilms were examined under a binocular and individual cyanobacterial morphotypes were transferred to plates containing solid Z8 medium [26] containing 0, 20, and 35 g/L salt (Instant Ocean, Aquarium Systems, France). Isolation in liquid Z8 medium was also attempted.

### 4.2. Strain Cultivation and Biomass Production

Back to the laboratory, surviving non-axenic strains were stabilized. Strains were registered in the Paris Museum Collection (PMC) under labels PMC 1050.18 to PMC 1092.19 (Table 1). They are maintained in liquid medium and are available upon request (https://www.mnhn.fr/fr/collections/ensembles-collections/ressources-biologiques-cellules-vivantes-cryoconservees/microalgues-cyanobacteries). Biomass was produced for two months in increasing volumes of liquid Z8 media (25 ± 1 °C; 15 μmol/m^2^/s white light; 16 h light: 8 h dark) without salt (PMC1069.18, 1070.18, 1071.18, 1072.18, 1073.18, 1092.18) or with 20 g/L salt (other strains). 

### 4.3. Strain Identification

DNA was extracted from cultures using the ZymoBIOMICS Fecal/Soil Kit (Zymo Research, Irvine, CA, USA) following manufacturer’s instructions including a 3 min disruption of cells using ceramic beads. Concentrations were measured using Nanodrop and Qubit. Fragments of the 16S rRNA-encoding gene were amplified by PCR for 35 cycles using two primer sets commonly used to specifically amplify cyanobacterial genes. Annealing temperature of 58 °C was used for primer set 8F (5’-AGAGTTTGATCCTGGCTCAG 3’) and 920R (5′-TTGTAAGGTTCTTCGCGTTG-3’), and annealing at 55 °C was used for primer set 861F (5′-TAACGCGTTAAGTATCCC-3′) and 1380R (5′-TAACGACTTCGGGCGTGACC-3′) [27,28]. For each strain, sequence chromatograms (Genoscreen, Lille, France) were examined, assembled using Geneious (https://www.geneious.com/), and compared to the GENBANK database using BLAST. Sequences are deposited in GENBANK under accession numbers MN823169 to MN823186; MN824246 and MN824247.

A dataset was built consisting of sequences from the 20 isolated strains, their best BLAST hits, and representatives of major cyanobacterial lineages. Sequences from genus *Gloeobacter* were used as an outgroup. Sequences were aligned using the secondary structure-aware Infernal Aligner v. 1.0 tool available on the Ribosomal Database Project website [29]. Alignment was controlled to remove ambiguously-aligned zones. Phylogenetic tree was reconstructed using the software MEGA7 [30]. Relationships were inferred using a Maximum Likelihood approach using a General Time Reversible model (5 categories and invariants), and 1280 nucleotide positions. Support values at nodes were obtained from 100 boostrap replicates analyzed using the same method. A pairwise p-distance matrix (Appendix A) was built to support preliminary genus and species delimitation.

### 4.4. Preparation of Cyanobacteria Chemical Extracts

An aliquot each culture was deposited in a 15 mL Falcon tube containing 10 mL of unsalted Z8 medium. After centrifugation at 4000 g and three successive washes with unsalted Z8 medium, the pellets were lyophilized under vacuum at −40 °C for 12 h. The dry extracts were weighed and then re-suspended in a MeOH/CH_3_CN/H_2_O mixture (40:40:20) for having a final concentration of 1 mg /100 μL of solvent mixture. After three successive sonications (6 min cycle: 1 min ON/30 s OFF) and centrifugations at 14,000 g for 5 min, the supernatants were evaporated and the dry extracts were prepared for having a final concentration of 1 mg/mL and were then filtered on a membrane of 13 mm in diameter and 0.2 μm in pores (VWR International). An aliquot of 30 μL was reserved for LC-MS² analysis. The remaining samples were evaporated and diluted in DMSO at a concentration of 10 μg/μL for antibacterial activities evaluation.

### 4.5. LC-MS^2^ Analyzes of Extracts

The extracts were subjected to an Agilent 1260 HPLC (Agilent Technologies, Les Ulis, France) coupled to an Agilent 6530 Q-ToF-MS equipped with a Dual ESI source. The chromatographic separation was performed using an HPLC (C18 Sunfire^®^ Waters 150 × 2.1 mm, 3.5 µm column, 250 µL/min gradient elution (A: CH_3_CN, B: H_2_O + 0.1% formic acid) from 5% to 100% A, over 20 min). The divert valve was set to waste for the first 3 min. In positive ion mode, purine C_5_H_4_N_4_ [M + H]^+^ ion (*m/z* 121.0509) and hexakis (1*H*, 1*H*, 3*H*-tetrafluoropropoxy) phosphazine C_18_H_18_F_24_N_3_O_6_P_3_ [M + H]^+^ ion (*m/z* 922.0098) (HP 0921) were used as internal lock masses. Source parameters were set as follow: capillary voltage at 3500 V, gas temperature at 320 °C, drying gas flow at 10 L/min, nebulizer pressure at 40 psi. Fragmentor was set at 175 V. Acquisition was performed in auto MS^2^ mode on the range *m/z* 100–1200 with an MS rate of 1 spectra/sec and an MS/MS scan rate of 3 spectra/sec. Isolation MS/MS width was 2 *m/z*. Fixed collision energies 45 eV was used. MS/MS events were performed on the three most intense precursor ions per cycle with a minimum intensity of 5000 counts. Full scans were acquired at a resolution of 11,000 [FWHM] (*m/z* 922). 

#### 4.5.1. MS/MS Data Pretreatment 

The MS data were converted from RAW (Thermo) standard data format to mzXML format using the MSConvert software, part of the ProteoWizard package [31]. The converted files were treated using the MZmine software suite v. 2.38 [12]. 

The parameters were adjusted as following: the centroid mass detector was used for mass detection with the noise level set to 1.0E6 for MS level set to 1, and to 0 for MS level set to 2. The ADAP [32] chromatogram builder was used and set to a minimum group size of scans of 2, minimum group intensity threshold of 3.0E3, minimum highest intensity of 3.0E3 and *m/z* tolerance of 10.0 ppm. For chromatogram deconvolution, the algorithm used was the wavelets (ADAP). The intensity window S/N was used as S/N estimator with a signal to noise ratio set at 10, a minimum feature height at 1000, a coefficient area threshold at 10, a peak duration range from 0.02 to 1.0 min and the RT wavelet range from 0.02 to 0.6 min. Isotopes were detected using the isotopes peaks grouper with a *m/z* tolerance of 10.0 ppm, a RT tolerance of 0.3 min (absolute), the maximum charge set at 1 and the representative isotope used was the most intense. Peak alignment was performed using the RANSAC alignment method (*m/z* tolerance at 10 ppm), RT tolerance 0.3 min, RT tolerance after correction 0.5 min, RANSAC iterations 0, Minimum number of points: 80.0 %, Threshold value: 0.3, requiring the same charge state. The peak list was gap-filled with the same RT and *m/z* range gap filler (*m/z* tolerance at 10 ppm). Eventually the resulting aligned peaklist was filtered using the peak-list rows filter option in order to keep only features associated with MS^2^ scans. 

#### 4.5.2. Molecular Networks Generation 

In order to keep the retention time, the exact mass information and to allow for the separation of isomers, a feature-based molecular network (https://ccms-ucsd.github.io/GNPSDocumentation/ featurebasedmolecularnetworking/) was created using the mgf file resulting from the MZmine pretreatment step detailed above. Spectral data was uploaded on the GNPS molecular networking platform. A network was then created where edges were filtered to have a cosine score above 0.7 and more than six matched peaks. Further edges between two nodes were kept in the network if and only if each of the nodes appeared in each other’s respective top 10 most similar nodes. The spectra in the network were then searched against GNPS’ spectral libraries. All matches kept between network spectra and library spectra were required to have a score above 0.7 and at least six matched peaks. The output was visualized using Cytoscape 3.6 software [33]. The GNPS job parameters and resulting data are available at the following address (https://gnps.ucsd.edu/ProteoSAFe/status.jsp?task=9581427a15b7422d8bd2b3b4b086189e). The DEREPLICATOR job resulting data is available at the following address (https://gnps.ucsd.edu/ProteoSAFe/status.jsp?task=0c058507ac774dd7b881c2ee36d57720).

### 4.6. Evaluation of the Antibacterial Activity of Cyanobacterial Strains

The antibacterial activities of the chemical extracts of the various cyanobacterial strains were tested against six human pathogenic bacteria (*Escherichia coli* ATCC 8739, *Klebsiella pneumoniae* ATCC 11296, *Pseudomonas aeruginosa* ATCC 13388, *Enterococcus faecalis* ATCC 29212, *Staphylococcus aureus* ATCC 6538 and *Bacillus cereus* ATCC 14579) and four marine pathogenic bacteria (*Vibrio alginolyticus* ATCC 17749, *Vibrio anguillarum* ATCC 19264, *Pseudoalteromonas atlantica* ATCC 19262 and *Pseudoalteromonas distincta* ATCC 700518). The selected pathogenic human and marine bacteria were cultured in LB (Luria Bertoni) medium at 37 °C or in MB (Marine Broth) at 25 °C, respectively. The different bacteria were isolated on LB or MB agar by incubation at 37 °C or 25 °C overnight. A pre-culture of 5 mL was prepared by inoculating a colony of each bacterial strain, and incubated at 37 °C or 25 °C and stirring overnight. The bacterial suspension was adjusted by dilution to obtain an optical density (OD) of 0.03 at 620 nm. The antibacterial assays were performed by a liquid method in 96-well microplates. Briefly, 100 µL of the bacterial suspension of different bacteria strains were distributed in each well. The extracts, diluted in DMSO, were tested in triplicate at a concentration of 100 μg/mL. The 96-well microplates were incubated overnight at 37 °C or 25 °C and shaked at 450 rpm. The OD of each well was measured at 620 nm using an absorbance reader plate (Multiscan, Thermofisher, Saint-Herblain, France). The percentage of growth inhibition was calculated using the formula: % inhibition = 100 − [(ODS − ODB)/(ODT − ODB) × 100] where T = bacterial suspension without test sample, B = culture medium without bacteria and S = bacterial suspension test sample. Standard antibiotics were used as positive controls (ampicillin against *E. faecalis*, *B. cereus, P. distincta*, *V. anguillarum*; chloramphenicol against *E. coli*, *P. aeruginosa*, *V. alginolyticus, P. atlantica*; gentamycin against *S. aureus*, *K. pneumoniae*).

## Figures and Tables

**Figure 1 marinedrugs-18-00016-f001:**
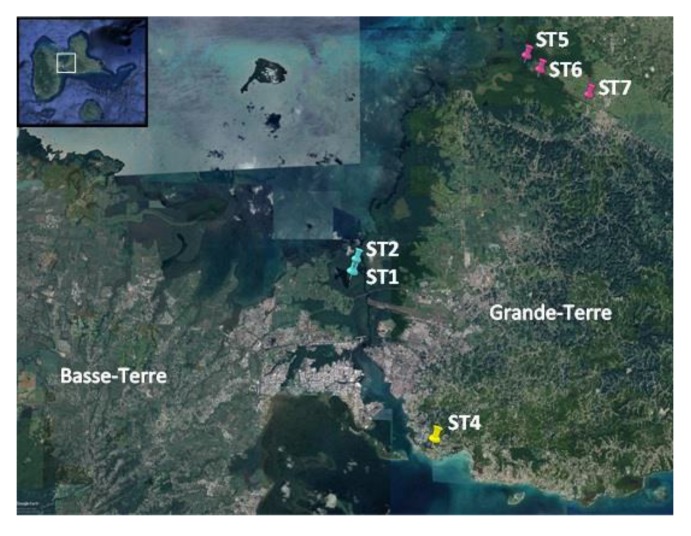
Google map showing the collection sites (ST: stations) located in Grande-Terre (ST4, ST5, ST6, and ST7) and in Basse-Terre (ST1 and ST2), the two main islands constituting Guadeloupe.

**Figure 2 marinedrugs-18-00016-f002:**
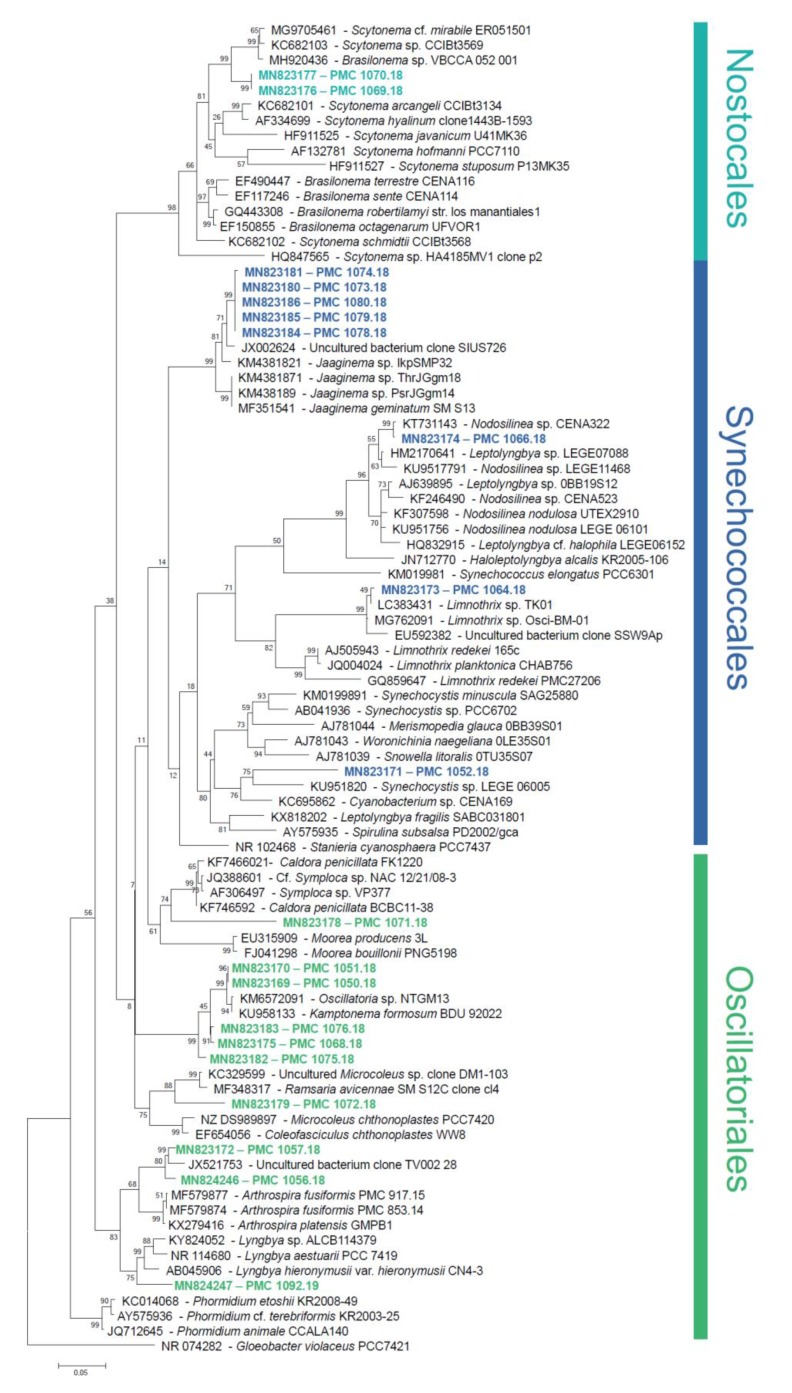
Phylogenetic tree based on the maximum-likelihood analysis of the bacterial 16S rRNA-encoding gene. Sequences from this study are in bold and colored. Support values at nodes were obtained from 100 boostrap replicates. Scale bar represents estimated 5% sequence difference.

**Figure 3 marinedrugs-18-00016-f003:**
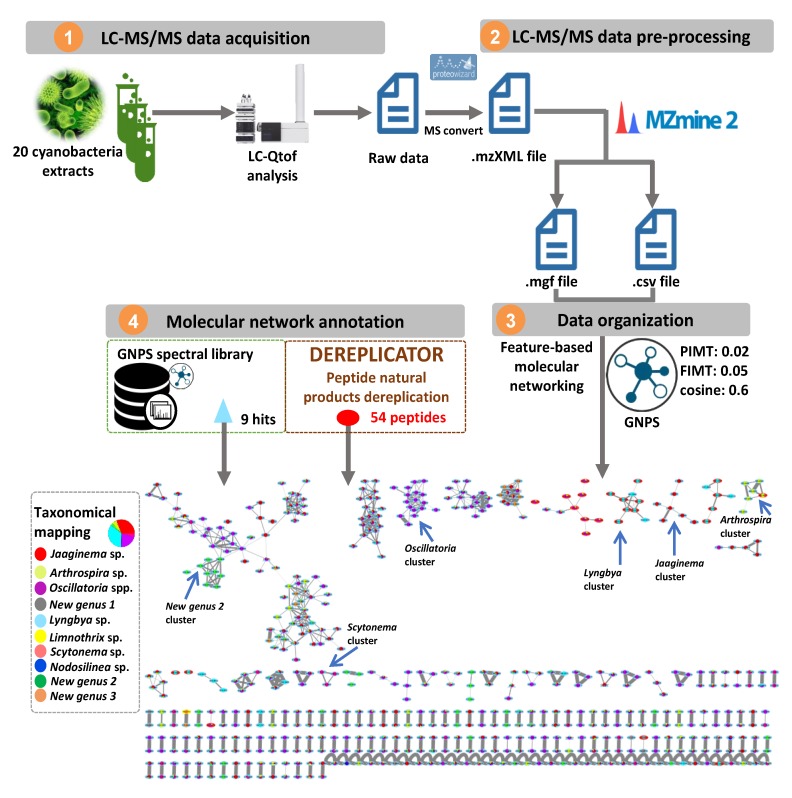
The molecular network obtained through the LC-MS/MS analysis of the 20 extracts of cyanobacterial strains collected in Guadeloupe. Peptides dereplicated via DEREPLICATOR are visualized in red ellipses. Nodes (round shape) are colored according to the mean precursor ion intensity per cyanobacterium genus (For further details See Appendix A).

**Table 1 marinedrugs-18-00016-t001:** Cyanobacterial strains isolated from Guadeloupe habitats (see map on Figure 1). Strain ID corresponds to the reference number in the Paris Museum Collection (PMC) of cyanobacteria from which strains are available upon request. Affiliation is according to the 16SrRNA-based phylogenetic analysis displayed on Figure 2 and to the distance matrix in Appendix A.

Strain ID	Order	Affiliation	Sampling Site	Coordinates	Isolation Source	Best BLAST Hit and % 16S rRNA Sequence Similarity
**PMC 1050.18**	Oscillatoriales	*Oscillatoria* n. sp. 3	Manche-à-Eau, ST1	16°16′32″ N/61°33′18″ W	Dense filamentous brown mat	*Oscillatoria/Kamptonema formosum* BDU 92022 (KU958133)/99%
**PMC 1051.18**	Oscillatoriales	*Oscillatoria* n. sp. 3			Dense filamentous brown mat	*Oscillatoria/Kamptonema formosum* BDU 92022 (KU958133)/99%
**PMC 1052.18**	Synechococcales	*Gen. Nov. 3*, n. sp. 1			Benthic mat with Cyanobacteria and *Beggiatoa*-like morphotypes	*Synechocystis* sp. (KU951820)/92%
**PMC 1056.18**	Oscillatoriales	*Arthrospira*, n. sp. 1	Manche-à-Eau, ST2	16°16′49″ N/61°33′13″ W	Benthic blue-green mat	Uncultured bacterium clone TV002_28 (JX521753)/98%
**PMC 1057.18**	Oscillatoriales	*Arthrospira*, n. sp. 1			Periphytic biofilm covering immersed roots of *Rhizophora mangle*	Uncultured bacterium clone TV002_28 (JX521753)/97%
**PMC 1064.18**	Synechococcales	*Limnothrix* n. sp. 1	Marina Bas du Fort, ST4	16°13′13″ N/61°31′24″ W	Dense green benthic biofilm containing Oscillatoriales, *Spirulina* and *Beggiatoa*-like morphotypes	*Limnothrix* sp. TK01 (LC383431)/99%
**PMC 1066.18**	Synechococcales	*Nodosilinea* n. sp. 1	Canal des Rotours, ST5	16°21′8.4″ N/61°29′20″ W	Dense periphytic biofilm covering immersed roots of *Rhizophora mangle*	*Nodosilinea* sp. CENA322 (KT731143)/99%
**PMC 1068.18**	Oscillatoriales	*Oscillatoria* n. sp. 2			Dense periphytic biofilm covering immersed roots of *Rhizophora mangle*	*Oscillatoria/Kamptonema formosum* BDU 92022 (KU958133)/98%
**PMC 1069.18**	Nostocales	*Scytonema* n. sp. 1			Dense periphytic biofilm covering immersed roots of *Rhizophora mangle*	*Scytonema* cf. mirabile ER0515.01 (MG970546)/96%
**PMC 1070.18**	Nostocales	*Scytonema* n. sp. 1			Dense periphytic biofilm covering immersed roots of *Rhizophora mangle*	*Scytonema* cf. mirabile ER0515.01 (MG970546)/96%
**PMC 1071.18**	Oscillatoriales	*Gen. Nov. 1*, n. sp. 1			Dense periphytic biofilm covering immersed roots of *Rhizophora mangle*	*Symploca* sp. NAC 12/21/08-3 (JQ388601)*/92%*
**PMC 1072.18**	Oscillatoriales	*Gen. Nov. 2*, n. sp. 1			Dense periphytic biofilm covering immersed roots of *Rhizophora mangle*	Uncultured bacterium clone DM1-166 (KC329581)/94%
**PMC 1073.18**	Synechococcales	*Jaaginema* n. sp. 1	Canal des Rotours, ST 6	16°20′50″ N/61°29′03″ W	Dense periphytic blue-green biofilm covering immersed roots of *Rhizophora nigra*	*Jaaginema* sp. PsrJGgm14 (KM438189)/98%
**PMC 1074.18**	Synechococcales	*Jaaginema* n. sp. 1			Dense periphytic blue-green biofilm covering immersed roots of *Rhizophora nigra*	*Jaaginema* sp. PsrJGgm14 (KM438189)/91%
**PMC 1075.18**	Oscillatoriales	*Oscillatoria* n. sp. 1			Dense periphytic blue-green biofilm covering immersed roots of *Rhizophora nigra*	*Oscillatoria/Kamptonema formosum* BDU 92022 (KU958133)/94%
**PMC 1076.18**	Oscillatoriales	*Oscillatoria* n. sp. 2	Canal des Rotours, ST7	16°20′19″ N/61°27′55″ W	Dense periphytic biofilm covering immersed branch fragment	*Oscillatoria/Kamptonema formosum* BDU 92022 (KU958133)/99%
**PMC 1078.18**	Synechococcales	*Jaaginema* n. sp. 1			Dense periphytic biofilm covering immersed branch fragment	*Jaaginema* sp. PsrJGgm14 (KM438189)/98%
**PMC 1079.18**	Synechococcales	*Jaaginema* n. sp. 1			Dense periphytic biofilm covering immersed branch fragment	*Jaaginema* sp. PsrJGgm14 (KM438189)/98%
**PMC 1080.18**	Synechococcales	*Jaaginema* n. sp. 1			Dense periphytic biofilm covering immersed branch fragment	*Jaaginema* sp. PsrJGgm14 (KM438189)/97%
**PMC 1092.19**	Oscillatoriales	*Lyngbya*, n. sp. 1			Dense periphytic biofilm covering immersed branch fragment	*Lyngbya* sp. ALCB114379 (KY824052)/98%

**Table 2 marinedrugs-18-00016-t002:** Summary of significant antibacterial activities among the 20 cyanobacterial strains expressed as percentage of growth inhibition against pathogenic human and marine environmental bacteria.

Strain ID	Gram-Negative Pathogenic Bacteria	
	Human *E. coli*	Environmental *P. atlantica*
***Oscillatoria* spp.**		
PMC 1051.18	100 ± 4.17	n.a.
PMC 1076.18	n.a.	50.87 ± 1.37
***Jaaginema* sp.**		
PMC 1078.18	n.a.	54.07 ± 8.50
PMC 1079.18	n.a.	60.95 ± 5.52
PMC 1080.18	n.a.	53.01 ± 4.42

n.a.: not active at the concentration of 100 µg/mL (i.e., inhibition below 50%); Chloramphenicol was used as positive control in both human *E. coli* and marine environmental *P. atlantica* pathogenic bacteria. Values presented as the mean ± SEM (*n* = 3).

**Table 3 marinedrugs-18-00016-t003:** Eleven peptide natural products (PNPs) previously isolated from cyanobacteria (in the increasing order of *P* values) dereplicated by DEREPLICATOR.

Variant PNP	Producer	Detected in	Score	*P*-Value	Variant PNP Mass	Peptide Mass *
Viequeamide A	*Marine Button* *Cyanobacterium* (unidentified)	*Oscillatoria* sp., *Scytonema* sp., *Arthrospira* sp.	12	2.8 10^−17^	803.50	892.55
Nostophycin	*Nostoc* 152	*Lyngbya* sp.	11	2.1 10^−15^	888.48	943.55
Wewakamide A	*Lyngbya semiplena* *Lyngbya majuscula*	*Oscillatoria* sp., *Scytonema* sp., *Arthrospira* sp.	12	4.4 10^−15^	994.65	891.59
Majusculamide C_Demethoxy	*Lyngbya majuscula*	*Oscillatoria* sp., *Scytonema* sp., *Arthrospira* sp.	11	4.6 10^−13^	954.58	892.49
Wewakazole	*Lyngbya majuscula*	*Lyngbya* sp.	12	6.7 10^−12^	1140.54	1058.54
Anacyclamide A10	*Anabaena* sp. *90*	*Lyngbya* sp.	10	9.6 10^−11^	1052.53	1009.49
Aerucyclamide C	*Microcystis* *aeruginosa*	*Lyngbya* sp.	7	9.5 10^−9^	516.22	645.28
Microcystin LR	*Microcystis bloom*	*Lyngbya* sp.	7	1.5 10^−8^	980.53	846.91
Pitipeptolide A	*Lyngbya majuscula*	*Oscillatoria* sp., *Scytonema* sp., *Arthrospira* sp.	7	1.6 10^−8^	807.48	795.51
Microcystin RA	*Microcystis*	*Lyngbya* sp.	7	1.7 10^−8^	952.50	825.38
Raocyclamide B	*Oscillatoria raoi*	*Arthrospira* sp.	6	6.5 10^−8^	568.21	581.17

*: Peptide mass (in Da) found in our extracts and previously isolated from Cyanobacteria.

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
