# Peer review of "New Benthic Cyanobacteria from Guadeloupe Mangroves as Producers of Antimicrobials"

_marinedrugs, 2019, doi:10.3390/md18010016_

Round 1

Reviewer 1 Report

This article discusses Benthic cyanobacteria strains from Guadeloupe and combines  phylogenetic, chemical and biological studies in order to better understand the taxonomic and chemical diversity as well as the biological activities of these cyanobacteria through the effect of their specialized metabolites.

This article should be published.

My only comment is that the structures of these metabolites should be studied in depth for future work on these organisms by using for example NMR spectroscopy.

Author Response

Reviewer 1:

Comments to the Author:

My only comment is that the structures of these metabolites should be studied in depth for future work on these organisms by using for example NMR spectroscopy.

Author’s reply: We agree with the reviewer that a deeper study of the selected cyanobacterial strains is recommended. We have begun the culture of some of them in this focus and we think that this paper is an exciting and promising result of exploration of benthic tropical cyanobacterial strains of French overseas, which, in contrast, with pelagic cyanobacterial strains have been little studied.

Reviewer 2 Report

This is an important manuscript exploring few Cyanobacteria diversity from Guadeloupe  and their ability to produce antimicrobial agents.

The work is very carefully designed and executed, and the results are also clear.

The authors isolated 20 new cyanobacterial strains, which represent 13 new species within three new and seven already known genera. Additionally, they found important antimicrobial activity against the human pathogenic bacterium E. coli. Certainly, the work has not yet reached the end, even from the taxonomic description or the identification of the active antimicrobial agents. However, this is the important initial step to investigate the benthic Cyanobacteria from tropical mangroves of Guadeloupe.

I propose to be accepted for publication in marine drugs with a minor modification.

A Google-map of Guadeloupe areas where samples were collected could be added, either in the last of page 3 or the page 4.

Author Response

Reviewer 2:

Comments to the Author:

A Google-map of Guadeloupe areas where samples were collected could be added, either in the last of page 3 or the page 4.

Author’s reply: As recommended, we have added a google map of Guadeloupe as well as the sites of collections and we have corrected some GPS mistakes in the Table 1.

Reviewer 3 Report

The paper from Duperron et al (MarineDrugs 666775) identified novel cyanobacterial species from Guadalupe, and analyzed their phylogeny, chemical diversity and anti-microbial activities.

Major comments:

The 9 identified peaks in GNPS should be described in the results and discussion section. A comparison is lacking between mass peaks from experimental data to data base matches (deviations of m/z). The EIC (extracted ion chromatogram) should be added for each peak in the figure S2. The corresponding revised version of the figure S2 could be added to the main text.

The masses of identified peptides in DEREPLICATOR compared to data base matches are quite different. Presented data are not convincing that identified masses correspond to the suggested peptides. Authors should at least discuss and highlight other papers that used a similar identification threshold to identify peptides from mass spectrometry in data bases.

Discussion should add on the definition of cutoff values for novel species and novel genera. Those thresholds should be discussed and compared to existing literature.

Discussion should add on the anti-microbial activity observed in the strains and compare the effective concentrations to other studies (concentrations) using cyanobacteria. The activity is high or low compared to other studies and to the targeted microbial species?

Author Response

Reviewer 3:

Comments to the Author:

The 9 identified peaks in GNPS should be described in the results and discussion section. A comparison is lacking between mass peaks from experimental data to data base matches (deviations of m/z). The EIC (extracted ion chromatogram) should be added for each peak in the figure S2. The corresponding revised version of the figure S2 could be added to the main text.

Author’s reply: We acknowledge the referee for this comment. The 9 hits identified in GNPS are not commonly described as cyanobacterial's specialized metabolites, that explains why we did not develop further this result. However, they are available through the link provided in the main text of the manuscript. (https://gnps.ucsd.edu/ProteoSAFe/result.jsp?task=9581427a15b7422d8bd2b3b4b086189e&view=view_all_annotations_DB)

The 11 identified peptide variants are now discussed in the discussion section. Instead of including figure S2 in the article, we propose to rather include a portion of Table S2 as Table 3 in the manuscript that will summarise the results of the DEREPLICATOR process regarding the 11 putatively indentified peptides. The deviations of m/z were not included in the table because they are too high and do not reflect the reliability of the identification since we consider analogs and not isomers.

The EIC (extracted ion chromatogram) should be added for each peak in the figure S2. The masses of identified peptides in DEREPLICATOR compared to data base matches are quite different. Presented data are not convincing that identified masses correspond to the suggested peptides. Authors should at least discuss and highlight other papers that used a similar identification threshold to identify peptides from mass spectrometry in data bases.

Author’s reply: We understand the referee’s concerns. Since the peptide natural products that have been dereplicated through DEREPLICATOR are absent in natural product spectral libraries, DEREPLICATOR proposed variants of known peptides with structural modifications. This process is known as variable dereplication. This statement has been added to the results section of the manuscript to further explain the tool.

Unfortunately, as the analogue peptides proposed by DEREPLICATOR are absent in the investigated cyanobacteria extracts, the EIC could not be performed.

Hereafter are summarized recent articles citing the DEREPICATOR algorithm :

- Martin H, C.; Ibáñez, R.; Nothias, L.-F.; Boya P, C. A.; Reinert, L. K.; Rollins-Smith, L. A.; Dorrestein, P. C.; Gutiérrez, M., Viscosin-like lipopeptides from frog skin bacteria inhibit Aspergillus fumigatus and Batrachochytrium dendrobatidis detected by imaging mass spectrometry and molecular networking. Scientific Reports 2019, 9, 3019.

-  Saurav, K.; Macho, M.; Kust, A.;Delawská, K.; Hájek, J.; Hrouzek, P. Antimicrobial activity and bioactive profiling of heterocytous cyanobacterial strains using MS/MS-based molecular networking. Folia Microbiol. 2019, 64, 645-654.

- Paulo, B. S.; Sigrist, R.; Angolini, C. F. F.; De Oliveira, L. G. New Cyclodepsipeptide Derivatives Revealed by Genome Mining and Molecular Networking. ChemistrySelect 2019, 4, 7785-7790.

These references have been added to the manuscript.

Discussion should add on the definition of cutoff values for novel species and novel genera. Those thresholds should be discussed and compared to existing literature.

Author’s reply: The retained cutoff values used in this study are based on recognized standards based on large scale comparisons and recommendation from reference works that attempted to identify biologically-relevant values. This is now clarified in the corresponding section of results that introduces the values (l.75): “Strains were affiliated to hypothetical species and genera based on widely accepted 16S rRNA similarity cutoff values for species and genus delimitation [7, 10], respectively, and monophyly with members of these species and genera. A large-scale comparison of 6,787 genomes from 22 prokaryotic phyla established that a 99% 16S rRNA similarity cutoff value should be retained to delimit species [10]; and reference taxonomic works on Cyanobacteria recommend a 95% cutoff of cyanobacterial genera delimitation [7]. »

Checking again the sequences, we realized that an error occurred with our affiliation of one sequence, namely PMC 1056. A few bases were indeed of low quality. We re-analyzed the sequence, and had to recalculate the distance matrix (supplementary table 1). The only change in results is that this sequence is now conspecific with PMC 1057, and thus we now have 12 species instead of 13. This has been changed throughout the manuscript (text, Table 1…).

Discussion should add on the anti-microbial activity observed in the strains and compare the effective concentrations to other studies (concentrations) using cyanobacteria. The activity is high or low compared to other studies and to the targeted microbial species?

Author’s reply: The inhibitory value of 100% of 1051 strain’s crude extract against E. coli at 100 μg/mL is an excellent result compared to all the other strains which did not shown any  activity at this concentration. This value is quite high for a crude extract. We did not detail this comment in the text as the concentration of 100 µg/mL is usually the maximum concentration tested, which should reveal activity. But we have added in the text that the result was obtained at the concentration of 100 µg/mL.

Regarding the inhibitory activity at 50% of P. atlantica strain, it is more moderate but significant.
